# Towards a "City in Nature": Evaluating the Cultural Ecosystem Services Approach Using Online Public Participation GIS to Support Urban Green Space Management

Yi Fan Koh [1], Ho Huu Loc [2,3,*] and Edward Park [1,4,*]

1  Humanities and Social Studies Education, National Institute of Education (NIE),
   Nanyang Technological University, Singapore 637616, Singapore; kohyifan96@gmail.com
2  Water Engineering and Management, School of Engineering and Technology (SET),
   Asian Institute of Technology, Pathum Thani 12120, Thailand
3  Department of Environment Management, Faculty of Food and Environment Management,
   Nguyen Tat Thanh University, Ho Chi Minh 70000, Vietnam
4  Earth Observatory of Singapore (EOS), Asian School of the Environment (ASE),
   Nanyang Technological University, Singapore 639798, Singapore
*  Correspondence: hohuuloc@ait.asia (H.H.L.); edward.park@nie.edu.sg (E.P.)

**Abstract:** The concept of cultural ecosystem services has been increasingly influential in both environmental research and policy decision making, such as for urban green spaces. However, its popular definitions tend to conflate "services" with "benefits", making it challenging for planners to employ them directly to manage urban green spaces. Thus, attempts have been made to redefine cultural ecosystem services as the function of cultural activities in environmental spaces which result in people's enjoyment of cultural ecosystem benefits. The operability of such a redefinition needs to be evaluated, which this study seeks to achieve with Bishan-Ang Mo Kio Park in Singapore presenting itself as a prime case study research area. Transdisciplinary mixed methods of a public participation geographic information system, which leverages on spatial data from public park users, and social media text mining analysis via Google reviews were used. A wealth of cultural ecosystem services and benefits were reported in the park, especially the recreational and aesthetic services and experiential benefits. Policy and methodological implications for future research and urban park developments were considered. Overall, this paper would recommend the employment of the redefined cultural ecosystem services approach to generate relational, data-driven and actionable insights to better support future urban green space management.

**Keywords:** cultural ecosystem services; urban green space management; Singapore; public participation geographic information system; social media text mining analysis

## 1. Introduction

Cultural ecosystem services (CES) have been an increasingly influential concept with established literature in environmental research and policy decision making [1–5], especially in urban green space (UGS) research and planning [5,6]. CES have been widely used in its popular definition as "the nonmaterial benefits people obtain from ecosystems" by the Millennium Ecosystem Assessment (MA) [7]. However, such a definition has proven to be problematic for urban planners to directly incorporate CES into decision making due to the conflation of the distinct terms of "services" and "benefits" [6,8].

Tandarić et al. (2020) thus proposed a more nuanced redefinition of CES to decouple "services" and "benefits" while highlighting their causal relationship, where the former will result in the latter (see Figure 1) [6]. Thus, CES are defined as the services that arise from cultural practices occurring in environmental spaces which allow people to enjoy the resultant cultural ecosystem benefits (CEB) that enhance their well-being [9]. CES may also be affected by cultural values held by people, such as how positive or negative they

perceive an ecosystem. Such decoupling has also been echoed by many researchers to better translate the CES concept into practice [6,10].

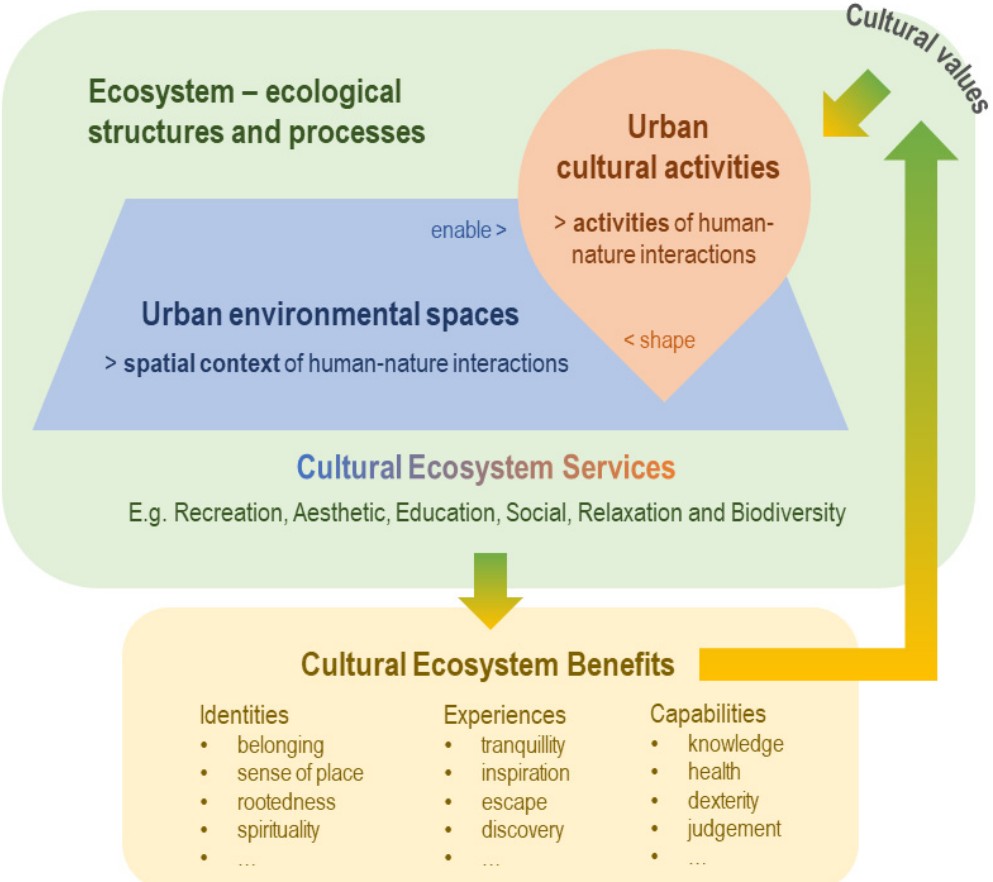

**Figure 1.** Redefined CES conceptual framework from Tandarić et al. (2020). Cultural ecosystem services are decoupled functions of human cultural activities taking place in environmental spaces. Its provision may be affected by cultural values held by people which will influence how they choose to interact with the ecosystem. Its presence may then generate cultural ecosystem benefits enjoyed by people. This may in turn influence the cultural values perceived and, in extension, the cultural ecosystem services offered in a continuous process.

With a long withstanding history of UGS planning implicitly underlain by the concept of ecosystem services [11], Singapore presents herself as a significant opportunity to further CES research into practice. The significance of Singapore is further augmented by the "City in Nature" vision which aims "to provide Singaporeans with a better quality of life (through engaging with CES and enjoyment of CEB), while co-existing with flora and fauna on this island" [12]. The new vision includes intensifying nature and greenery in new and existing local parks with the incorporation of natural designs, plantings, naturalisation of water features and conservation of important biodiversity [12]. These developments are evident in Bishan-Ang Mo Kio Park (BAMKP), with its concrete canals transformed into natural rivers [12] alongside its wide array of flora and fauna [13]. BAMKP has also been slated for further rewilding, with more naturalistic landscapes that promote the thriving of biodiversity [14].

Using BAMKP as a case study research area, this paper seeks to evaluate the operability of Tandarić et al.'s (2020) redefined CES conceptual framework to support UGS management, specifically with regard to Singapore's "City in Nature" vision. This paper thus proposes the following four research questions:

1. What are the public's cultural values of Bishan-Ang Mo Kio Park?

2.     What are the cultural ecosystem services offered in Bishan-Ang Mo Kio Park?

3.     What are the cultural ecosystem benefits enjoyed in Bishan-Ang Mo Kio Park?

4.     What is the relationship between services and benefits in Bishan-Ang Mo Kio Park?

Moreover, the contextual significance of Singapore gearing towards the new "City in Nature" vision positions this paper in a timely manner. Planners in Singapore may be supported with insights for future development and re-naturalisation of UGS which also complement the Natural Capital Singapore project spearheaded by the Singapore government to "assess the current status and health of Singapore's major ecosystems and quantify their economic and societal value" [15]. This paper also seeks to bridge the geographical gap in CES literature. Except for China, few Asian countries have been publishing CES research in quantity comparable to the West [16]. Besides, current CES research in tropical Asia tends to focus on the rural environments [17] despite the region's rapid urbanisation which heightens the importance of urban CES research in Asia [9,18–21]. Therefore, findings from this paper could also provide deep insights to support UGS planning in the other urbanised and urbanising tropical Asian countries.

## 2. Research Area

Located central of Singapore within Bishan town and flanked by Ang Mo Kio town on its northern edge, BAMKP (see Figure 2) was first constructed in 1988 as a 52 ha Bishan Park with a separate 2.7-kilometre-long concrete canal [22]. It underwent a revamp between 2009 and 2011 which overhauled the entire park with significant enhancements to park features, nature and biodiversity [22]. The reopened BAMKP in 2012 is one of Singapore's largest urban parks spanning an increased area of 62 ha with a 3-kilometre-long naturalised and meandering Kallang River. Most blue water features concentrate towards the south, whereas the northern region is mainly green with trees and plants. BAMKP is also conveniently bisected by a traffic road into two smaller and distinctly featured parks, with the River Plains in the east and Pond Gardens in the west (refer to Supporting Information Figure S1).

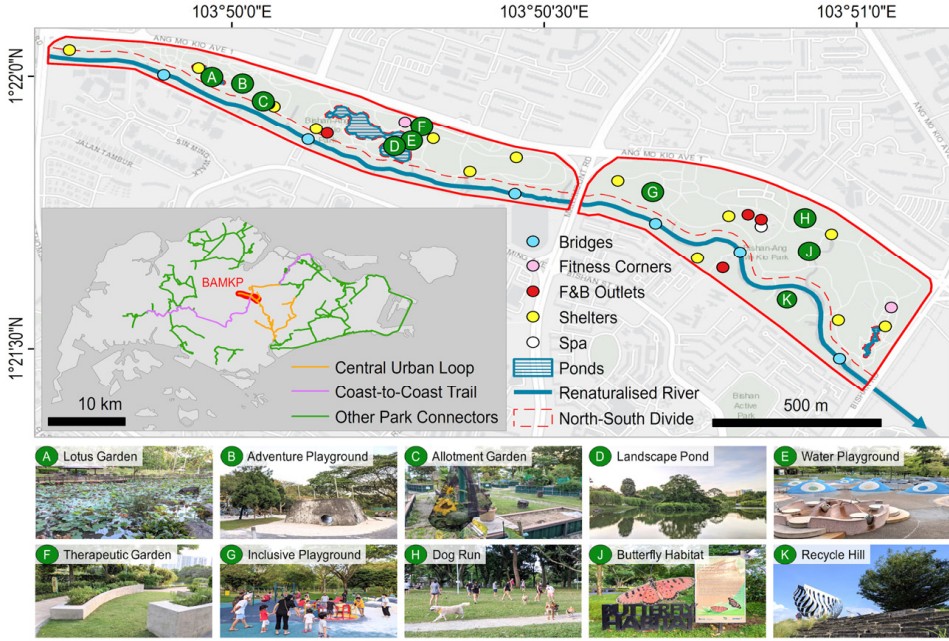

**Figure 2.** Research area of Bishan-Ang Mo Kio Park with its key park features. The red dashed line divides the park into the greener and more vegetated north and blue-green south richer in water features, such as the renaturalised river and ponds. The inset map shows the central location of the park amidst the network of park connectors in Singapore.

While the regional park predominantly serves residents in its immediate neighbourhood, it remains attractive for Singaporeans residing further away [22]. BAMKP, with almost 6 million visitors per year, is ousting Singapore's globally famed Gardens by the Bay (5.1 million visitors per year) and the Botanical Gardens (4.4 million visitors per year). Despite its relatively smaller park size, BAMKP has almost twice the number of visitors per hectare per year as compared to the other two popular parks [22]. The park is also plugged into Singapore's extensive and growing nation-wide Park Connector Network (PCN) that links different green spaces in Singapore together [23]. As part of the Central Urban Loop, BAMKP serves as a central node of the latest Coast-to-Coast (C2C) Trail that spans diagonally across the island from the west to the northeast [24].

## 3. Methodology

### 3.1. Differentiating Services and Benefits Using Mixed Methods

To evaluate the operability of Tandarić et al.'s (2020) redefined CES conceptual framework and support Singapore's "City in Nature" vision, the nuanced framework (see Figure 1) would serve as the backbone of the research methodology in this study [6]. Instead of relying on a singular methodology, mixed methods have been deemed more advantageous in exposing truer cultural valuations of places [25]. Thus, the online public participation geographic information system (PPGIS) surveys served as the primary method to uncover the spatial distribution and cultural values of CES alongside the enjoyment of CEB in BAMKP. Such decoupling of CES and CEB in PPGIS studies have and would further augment the credibility and usefulness of the result findings for planning and decision making [25].

To supplement the PPGIS survey findings, social media text mining analysis through Google reviews of BAMKP served as the secondary mode of analysis. The analysis of public reviews will highlight the relative presence of CES and CEB in BAMKP alongside cultural values via sentiment analysis. While analysis of social media content in CES research has been popularly performed through photographs, texts have also been emerging as another unique medium [26–28]. Social media text mining analysis method is advantageous as a rapid and cost-effective assessment of CES [29,30]. Both methods thus seek to complement each other to uncover a truer reflection of the CES, CEB and cultural values present in BAMKP.

### 3.2. Public Participation Geographic Information System (PPGIS)

#### 3.2.1. Data Collection

PPGIS was conducted using a 20-question online survey via Map-Me [31] and Google Forms (refer to Supporting Information Method S1). The questionnaire survey was developed based on references made to previous studies related to quantifying and mapping CES, see for instance, Yee et al. (2021) [4] or Loc et al. (2021) [5]. The questionnaire was pre-tested at the site and further improved, especially in terms of wording. The final version of the questionnaire used in this study is included as the Supplementary Materials. In addition, the participatory mapping method using Map-me was advantageous in obtaining truer and ground sentiments of actual park users' usage and enjoyment of the park alongside their mapped place values [32]. Map-Me was chosen as the ideal platform for its public accessibility and "spray can" function to better preserve the respondents' spatial responses which tend not to be reduced to simply points, lines and polygons [1–3,31]. One hundred valid participants, who were verified to be at least 21 years old and have ever visited BAMKP, were recruited from October to November 2020. Convenience snowball sampling was employed where participants were reached via word of mouth through email and social media platform, starting and extending from the acquaintances of the researchers. For the spatial questions on Map-Me, participants responded by mapping out blobs on a curated map. There is no limit in the number of blobs afforded to the participants, to encourage truer and fuller responses to the spatial questions. Ground truthing was also carried out in September 2020 to ensure that the key park features were still present and georeferenced

accurately on the Map-Me survey map (refer to Supporting Information Figure S1). A total of six CES and three CEB that were relevant and of interest to BAMKP were selected for the survey (see Table 1) based on both Tandarić et al.'s (2020) CES conceptual framework and previously conducted studies [13,22].

**Table 1.** List of the selected CES and CEB relevant for this study and some of their related words for text mining analysis.

| | Categories | Description | Word Examples |
|---|---|---|---|
| **Cultural Ecosystem Services (CES)** | Recreation | Park allows enjoyment of recreational activities. | playground, cycle, dog |
| | Aesthetic | Park allows admiration of natural or man-made beauty. | beautiful, scenic, clean |
| | Education | Park allows learning about nature and environment. | learn, sciences, guide |
| | Social | Park allows socialisation with people, family and friends. | picnic, gathering, kids |
| | Relaxation | Park allows relaxation and unwinding from everyday life. | relax, stroll, refreshing |
| | Biodiversity | Park allows getting near or watching animals and plants. | trees, otters, kingfisher |
| **Cultural Ecosystem Benefits (CEB)** | Identities | Park is beneficial in shaping personal identities. | spiritual, personal, past |
| | Experiences | Park is beneficial in providing positive experiences. | fun, social, enjoyable |
| | Capabilities | Park is beneficial in enhancing capabilities of people. | health, career, learning |

### 3.2.2. Data Analysis

For data visualisation and interpretation, descriptive statistics was used to highlight the data spread with graphical representations such as bar and lollipop charts. Inferential statistics was performed through one-way ANOVA tests to identify statistically significant differences. For the spatial data collected, CES heat maps were created using the Point Density geoprocessing tool from ArcMap 10.4 for hotspot analysis where the concentration of CES was identified and further analysed. Using the CES heat maps with equal intervals, spatial correlation using the Jaccard's Index [33] was also calculated by dividing the CES-mapped area over the total land area to determine the level of spatial correlation between each CES.

CES densities in the characteristically different north–south and east–west regions of BAMKP were also calculated to expose possible regional differences. As the blue water features predominate the southern region, the north–south segregation uncovers the effectiveness of these blue water features in augmenting CES presence in BAMKP. The east–west segregation follows the existing division of BAMKP into the western Pond Gardens and eastern River Plains [13]. Heat maps of CES spatial distribution relating to a high degree of CEB enjoyment were also generated. Using the CES mapped blobs of participants who have reported a high score of 4.5 for each CEB, three CES-CEB point density heat maps were generated for each CEB. The score of 4.5 was an optimal threshold based on a sensitivity analysis conducted to ensure the generation of meaningful heat maps for analysis (see Supporting Information Figure S2).

### 3.3. Social Media Text Mining Analysis

#### 3.3.1. Data Collection

Google reviews was chosen as the suitable social media platform for text mining analysis to be carried out in this study due to its sheer quantity of textual public opinions of BAMKP incomparable with other social media platforms (*n* = 5762 reviews as of

17 November 2020). All the textual Google reviews of BAMKP until 17 November 2020 were first extracted using ScrapeHero (https://www.scrapehero.com/marketplace/google-reviews/ accessed on 20 November 2020) before the data were cleaned and processed using the R software [34]. Reviews with non-English or no textual reviews were first removed before the remaining review sentences were split into individual words. Misspelled, non-English and less meaningful stop words were then removed, resulting in 2356 unique words for further analysis.

### 3.3.2. Data Analysis

The resultant unique words were then analysed through occurrence frequency analysis and sentiment analysis, where the former showed the degree of presence of CES and CEB while the latter showed the cultural values attached to BAMKP. Occurrence frequency analysis was conducted by associating the same six CES and three CEB of interest with the resultant unique words before conducting a word frequency count. The classification indicators for the six CES and three CEB was established based on the following definitional criteria (see Table 1). For example, the word "playground" was associated with recreation CES while the word "spiritual" was associated with identities CEB. Words may be associated with both a CES and a CEB if deemed suitable, such as "learn" and "learning" which were both associated with education CES and capabilities CEB at the same time. Sentiment analysis of the cultural values was also performed using the R software [34] with the lexicons "AFINN" [35] and "NRC" [36] to expose how positively or negatively the words are associated as well as the emotions captured in Google reviews, respectively.

## 4. Results

### 4.1. Sociodemographic and Park Usage Information

Of the 100 valid participants who completed the online PPGIS survey, most were aged 21–30 years old (81%) and of Chinese ethnicity (89%). This is likely due to the digital focus of the survey and its promotion via word of mouth on social media platforms among the younger and predominantly Chinese demographics. Most participants have identified themselves as irreligious, Buddhists or Christians (83%) and have attained a relatively high education level of a bachelor's degree, junior college certificate or postgraduate degree (88%). Gender-wise, there were also slightly more female (59%) than male (41%) participants (refer to Supporting Information Figure S3).

Regarding park usage patterns (see Figure 3), most participants have recently visited the park within the past year (70%), but only 13% have visited the park weekly in the same period. Participants generally spent an average of 30–60 min in BAMKP (49%) and with their friends (47%). The participants' residential proximities to BAMKP, as proxied by their postal codes in relation to Singapore's postal district zones [37], were found to be spatially distributed in a relatively balanced manner.

### 4.2. What Are the Public's Cultural Values of Bishan-Ang Mo Kio Park?

Generally, findings from the sentiment analysis of Google reviews have yielded that the public's cultural values towards the park were highly positive (see Figure 4). Using the "AFINN" lexicon in R [35], 93.9% of the review words were associated with positive connotations, with 58.9% of them valued at least a +3 or above. Using "NRC" as another lexicon in R that performs sentiment analysis more emotively [36], 84.8% of the review words were associated with positive emotions such as joy, trust, anticipation, and surprise, with the emotion of joy being the most significant (31.7%).

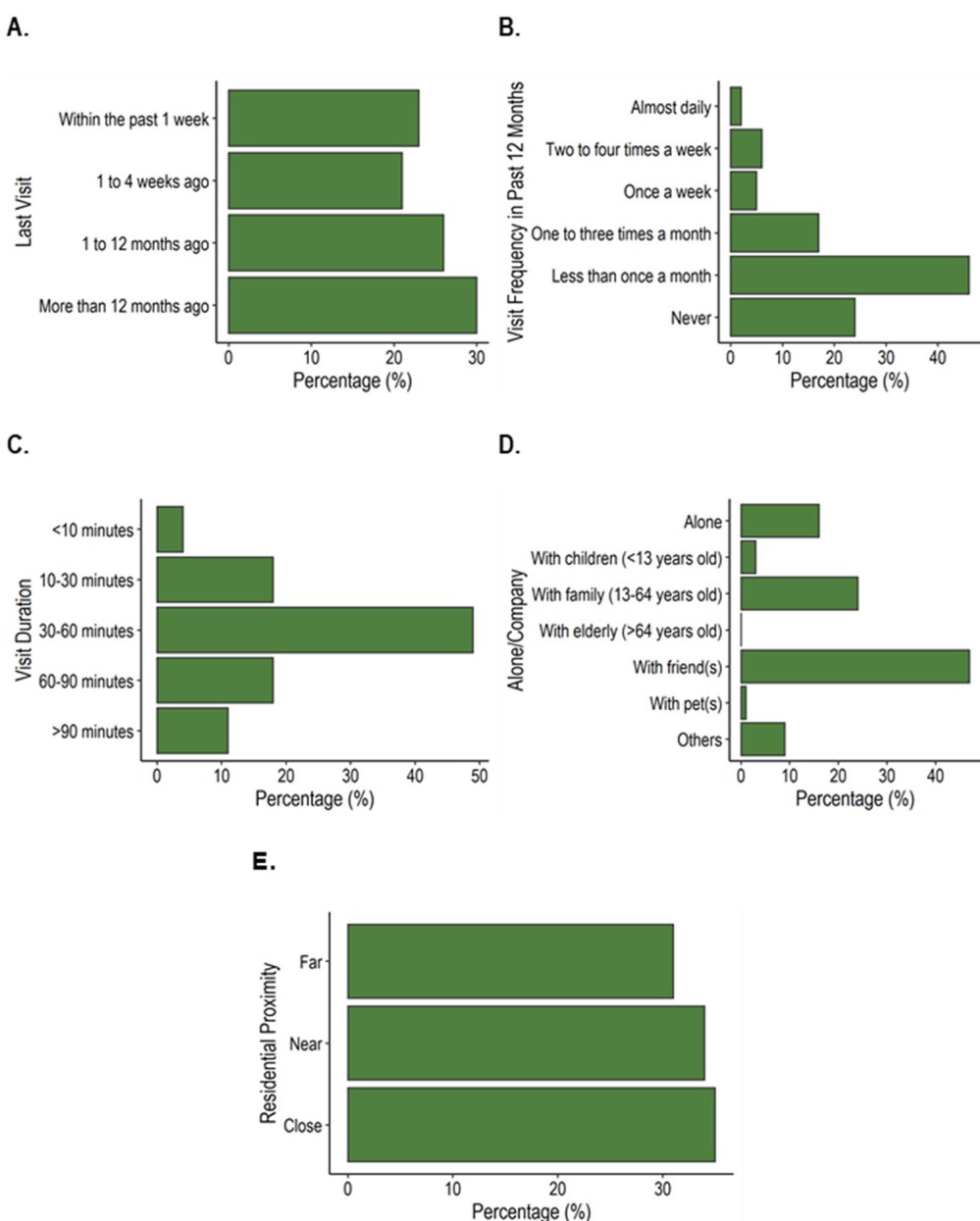

**Figure 3.** Park usage patterns of PPGIS participants. (**A**) Graph of participants' last park visit. (**B**) Graph of participants' park visit frequency in the past 12 months. (**C**) Graph of participants' average park visit duration. (**D**) Graph of participants' usual accompaniment for park visits. (**E**) Graph of participants' residential proximity to the park.

PPGIS surveys found that the park features planned and present in BAMKP were generally valued as important (see Figure 5). The common and non-exclusive park features such as the bridges (93% importance), shelters (93%), fitness corners (89%) and food and beverages (F&B) outlets (87%) were the most highly valued, likely for their widespread utility. The iconic naturalised river meandering through BAMKP was also among the most highly valued (92%). The other ten more localised and specific park features found in either the Pond Gardens or River Plains were deemed of secondary and moderate importance (62–77%).

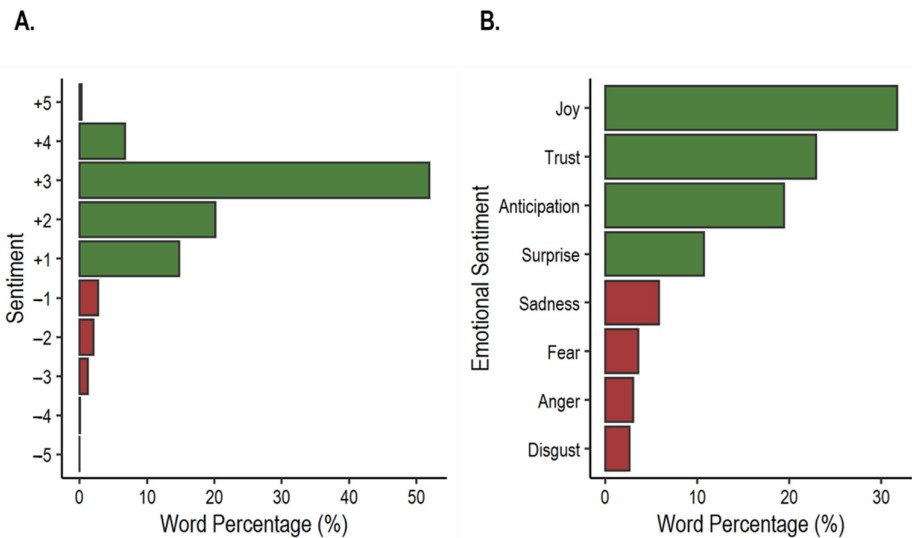

**Figure 4.** Cultural values of the entire BAMKP. (**A**) Sentiment analysis of Google review words ranging from the most emotively positive (+5) to the most emotively negative (−5) words. (**B**) Emotional sentiment analysis of Google review words.

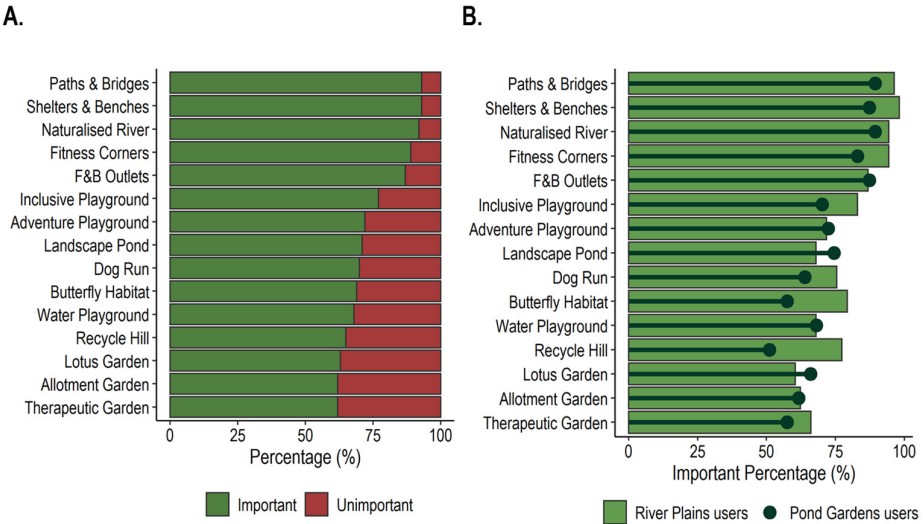

**Figure 5.** Cultural values of specific environmental spaces in BAMKP. (**A**) Degree of importance and unimportance of park features by PPGIS participants. (**B**) Degree of importance of park features by PPGIS participants according to their dominant park usage.

The cultural values of these specific environmental spaces were also uncovered to be influenced by the park users' dominant park usage. For example, less Pond Gardens users found the Recycle Hill located in the River Plains important (51%), whereas more River Plains users acknowledged its value (77%). It was also found that park users who dominantly frequented the River Plains generally reported higher valuations for all the BAMKP features.

When rescaling the analysis to inspect more closely on the cultural values of the CES, their importance has been generally well valued based on the PPGIS survey findings (see Figure 6). Using a Likert scale, all the CES in BAMKP scored a high average of 4.25 and ranged relatively high between 3.77 and 4.56. Recreation (4.56) and relaxation (4.44) CES were the most highly valued followed by biodiversity (4.29), social (4.25) and aesthetic (4.21) CS. The least culturally valued CES was found to be related to education (3.77), likely since BAMKP was primarily designed as a recreational park rather than an educational one [13].

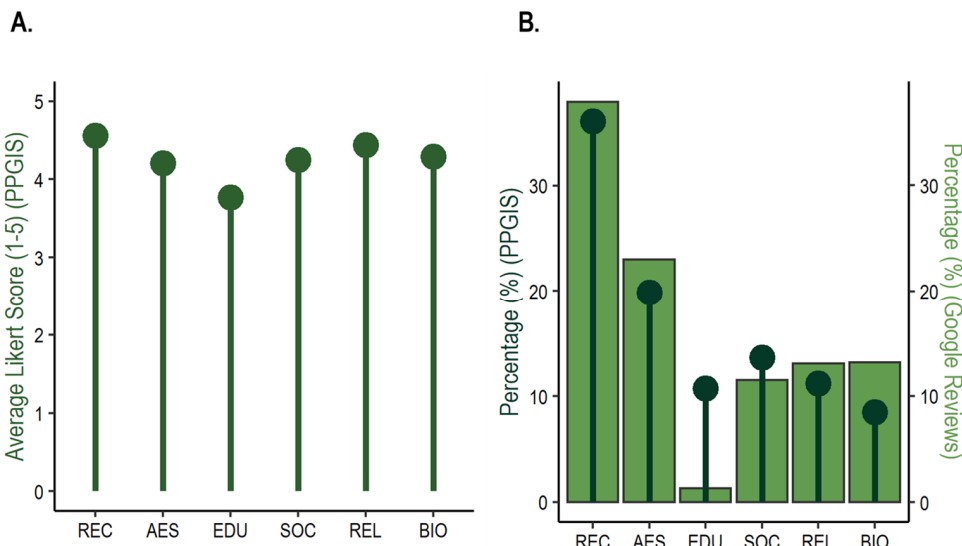

**Figure 6.** Cultural values and presence of CES in BAMKP. (**A**) Cultural values of CES in BAMKP via a Likert scale, where 1 and 5 represent the utmost unimportance and importance of CES to PPGIS participants respectively. (**B**) Presence of CES in BAMKP via percentage of blobs mapped by PPGIS participants (left axis) and percentage of CES-related words from Google reviews (right axis). Abbreviations: REC, recreation; AES, aesthetic; EDU, education; SOC, social; REL, relaxation; BIO, biodiversity.

*4.3. What Are the Cultural Ecosystem Services Offered in Bishan-Ang Mo Kio Park?*

4.3.1. Relative Presence

Although the PPGIS participants have placed considerable importance on all the CES offered in BAMKP, their engagement with the CES was more varied (see Figure 6). Using the number of blobs mapped by all participants for each respective CES as a proxy, the most offered and engaged CES in BAMKP was recreation (36.1%) and aesthetic (19.8%). In the word occurrence frequency analysis of Google reviews, recreation (37.9%) and aesthetic (23%) were also the top two CES, likely due to its "beautiful 3 km meandering river with lush banks of wildflowers" which makes "it a popular choice for nearby residents for recreational activities" [13]. Popular recreational activities in BAMKP include walking, running, cycling, yoga and even Chinese martial arts [22]. Recreation has also been routinely and widely recognised as the dominant CES in many other UGS [38].

From the PPGIS survey, except for the anomalous education CES (1.3%), findings from the word occurrence frequency of Google reviews also shed a similar light for social, relaxation and biodiversity CES (11.5–13.2%). However, education CES are likely still present in BAMKP as reported by the PPGIS participants (10.7%), where some of them have visited the park "with students/colleagues" as teachers or "with classmates" as students.

4.3.2. Spatial Distribution

Based on the PPGIS mapping of all six CES, CES heat maps were curated to show the extensiveness and concentration of CES present in BAMKP (see Figure 7). Generally, it was found that all the CES in BAMKP were well spread across the entire spatial extent of the park. This is likely due to both the high utilisation of park spaces with amenities and the extensive network of pathways present throughout BAMKP. The localised CES hotspots also tend to be located near the renaturalised water features, as these famed spots in BAMKP tend to have higher levels of social activities [22]. More specifically, the hotspots coincided with prominent park features such as the McDonald's F&B outlet, Recycle Hill and Therapeutic Garden, whereas coldspots with an absence of CES congregated near less accessible areas with fewer amenities.

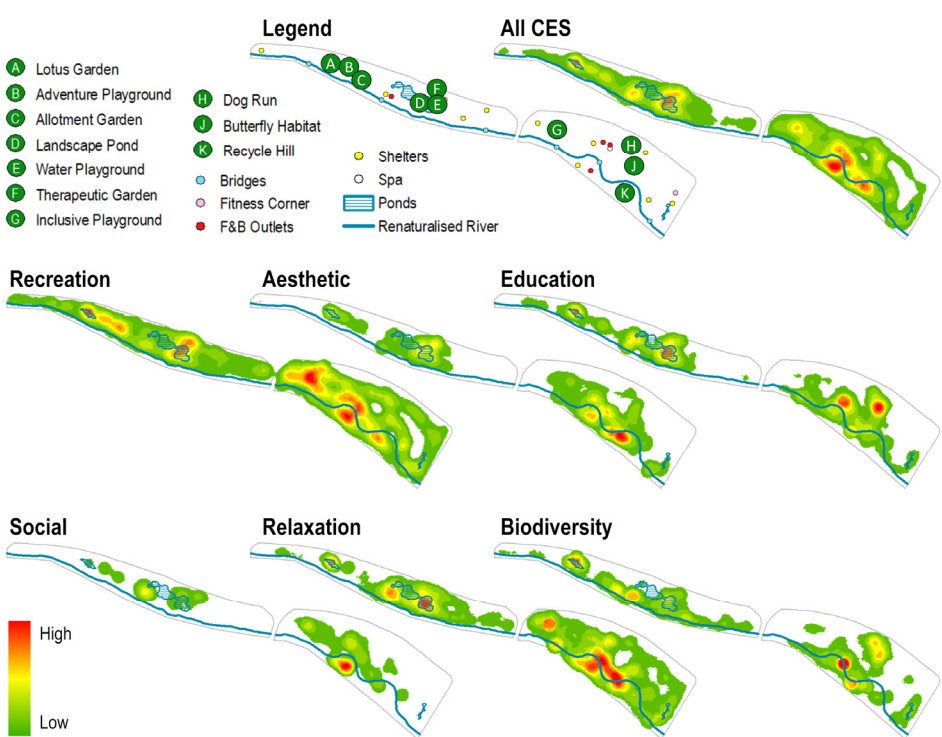

**Figure 7.** Spatial distribution of cultural ecosystem services in Bishan-Ang Mo Kio Park.

Recreation and relaxation CES were found to be the most widespread throughout the park due to their requirement for participants to be moving about. As aesthetic, education, social and biodiversity CES tend to be offered in a more static manner, the spread of these CES throughout the park were only moderately expansive. Social CES are the least widespread and only covered the central spaces of both bisected parks. This is likely since socialisation is more likely to take place in a highly static and preferably sitting environment, such as at shelters or F&B outlets.

The locations of CES hotspots for each specific CES were also generally situated near renaturalised water features and popular park features, such as the McDonald's F&B outlet (recreation, aesthetic, social and relaxation CES), Recycle Hill (recreation, aesthetic and relaxation CES) and Landscape Pond (recreation, aesthetic, education, relaxation CES). Anomalous and unique hotspots were also present for each CES, such as the two playgrounds and the community Allotment Garden for recreational CES. Distinct hotspots were also identified for education CES located near the Butterfly Habitat and a nearby shelter, likely due to the presence of informative displays and signages. As for biodiversity CES, three notable anomalous hotspots were found at the bridge near the McDonald's F&B outlet, Butterfly Habitat and Dog Run. The prime location of this bridge is near the popular McDonald's F&B outlet which also offers sightings of wildlife such as the Purple Heron [39].

### 4.3.3. Regional Distribution of Cultural Ecosystem Services Density

Given that there are moderate differences in park characteristics along the north–south and east–west transects of the park, regional analysis of CES density was performed for these four regions (see Figure 2 and Table 2). The north–south transect differs by their proportion of blue-water and green-vegetated features, with both features dominating the southern area while the northern area only offered the green-vegetated features. For this study, the southern BAMKP is defined to include areas south of 30m north of the river and within a 5m buffer around ponds while northern BAMKP consists of the remaining areas (see Figure 2). Conversely, the east–west bisection of the park into the eastern River Plains

and western Pond Gardens was officially designed by park planners to have distinct and unique park features in each bisect of the park.

**Table 2.** Regional distribution of CES density in BAMKP.

| | CES Density (Blobs/ha) | | | | | | |
|---|---|---|---|---|---|---|---|
| | Recreation | Aesthetic | Education | Social | Relaxation | Biodiversity | All CES |
| North (Green) | 873 | 370 | 215 | 318 | 232 | 155 | 2162 |
| South (Blue-Green) | 795 | 645 | 319 | 325 | 320 | 283 | 2687 |
| East (River Plains) | 906 | 503 | 243 | 370 | 295 | 187 | 2504 |
| West (Pond Gardens) | 767 | 415 | 263 | 254 | 220 | 216 | 2135 |

Generally, it was found that the blue-green south of BAMKP has about 24% higher CES density than the predominantly green north. Most of the CES, except for recreation, has higher CES density in the southern blue-green part of the park than the green north. The southern dominance of CES density was the greatest for biodiversity (83% higher) and aesthetic (74% higher) and least for social CES (2% higher). Such a concentration of high CES density may signify the success of the BAMKP's renaturalisation programme. Recreation CES was the only anomaly where the northern green region is denser in CES than the blue-green south (10% higher) likely due to more suitable environmental spaces, such as the pathways for jogging and cycling, the two playgrounds and the community Allotment Garden.

The results from regional analysis of the two smaller bisected parks indicated that the eastern River Plains tend to possess higher CES density than the western Pond Gardens, at about 17% denser. Similarly, the majority of the CES was denser in presence in the eastern rather than the western park. While the proportionate differences between the east–west regions were not as significant as that between the north–south regions, social (46% higher) and relaxation CES (34% higher) had the greatest regional disparity. The dominance of CES density in the eastern River Plains could be due to the higher number of park visitors as compared to the western Pond Gardens [22], likely because of higher population density surrounding the eastern River Plains and proximity to Bishan metro station [22]. The two anomalous CES with higher density in presence in the western River Plains were education and biodiversity CES which suggests that the park features there may have more potential in gearing BAMKP towards the "City in Nature" vision.

### 4.3.4. Spatial Correlations

The CES spatial correlation was calculated using Jaccard's Index [33] to determine the degree of spatial correlation between each of the CES (see Figure 8). Overall, as this research leans towards the social sciences, there is a moderately high degree of spatial correlation (coefficient value of 0.3 to 0.5) between all the CES pairings, except some biodiversity CES. The most spatially similar pairing was found to be between recreation and aesthetic CES (0.49), followed by aesthetic–social (0.39), recreation–relaxation (0.38) and aesthetic–education (0.38). Generally, pairings with recreational and aesthetic CES yielded more spatially similar correlations. Conversely, pairings with biodiversity CES generated the least spatially similar correlations, with biodiversity–social (0.25), biodiversity–relaxation (0.26) and biodiversity–education (0.28).

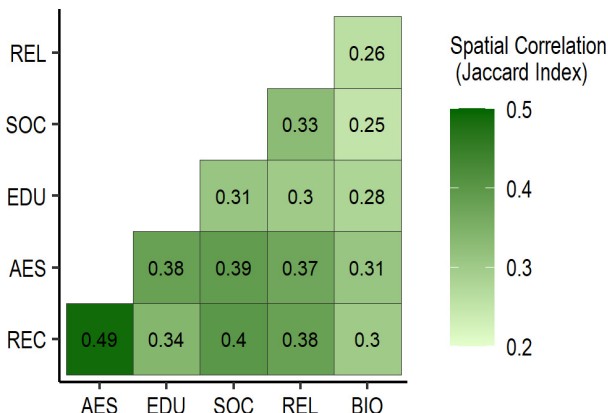

**Figure 8.** Spatial distribution of CES in BAMKP. Abbreviations: REC, recreation; AES, aesthetic; EDU, education; SOC, social; REL, relaxation; BIO, biodiversity.

*4.4. What Are the Cultural Ecosystem Benefits Enjoyed in Bishan-Ang Mo Kio Park?*

From the PPGIS survey, participants generally reported a moderately high degree of CEB presence with an average score of 3.60 on a Likert scale (see Figure 9). The experiential benefits were the most prevalent (3.87), followed by capabilities (3.66) and identities benefits (3.27). Similar findings were obtained from the occurrence frequency analysis of Google review words where most CEB-related words were associated with experiential benefits (81.1%). This is likely due to the intentional designed of the park for experiential activities, such as exercising, playing, wildlife watching and photography [13]. The one-way ANOVA test yielded no statistically significant differences for the experiential benefits, suggesting the general universality of the experiential enjoyment that park users may benefit from BAMKP (refer to Supporting Information Table S1).

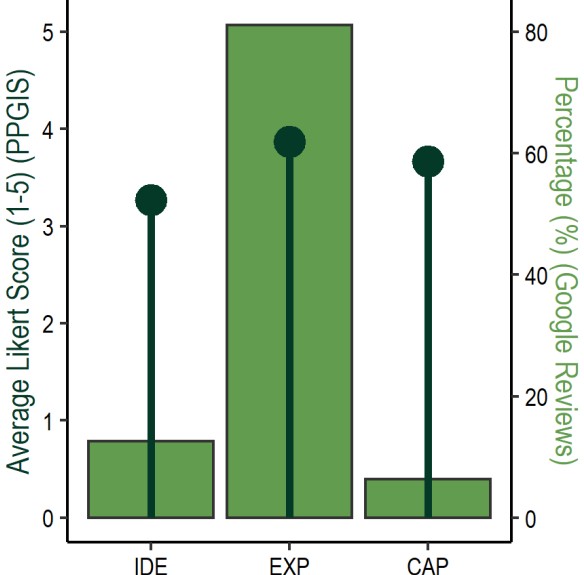

**Figure 9.** Presence of CEB in BAMKP via a Likert scale where 1 and 5 represent the utmost absence and presence of CEB enjoyed by PPGIS participants respectively (left axis) and percentage of CEB-related words from Google reviews (right axis). Abbreviations: IDE, identities; EXP, experiences; CAP, capabilities.

Conversely, the relatively lower presence of identities and capabilities CEB may be attributed to how these two types of benefits may require more sustained and repeated park visits (see Figure 9). This is supported by findings from the one-way ANOVA test where residential proximity and park visit frequency were found to play a statistically significant

role in one's identity CEB. People who lived near to the park and have visited the park frequently tend to report higher levels of benefits to their identity formation. However, 70% of the PPGIS participants have only visited BAMKP less than once a month or not at all in the past year. Such infrequent park visits could lead to the relatively lower identity and capabilities CEB being reported. Besides, Singapore's cityscape is generally blanketed with a variety of trees, shrubs and grasslands and everyday exposure to them could already have provided for the identity and capabilities CEB, such as improvement of one's mental health [40] and thus weakened people's association of these CEB in BAMKP.

### 4.5. What Is the Relationship between Services and Benefits in Bishan-Ang Mo Kio Park?

An understanding of the relationship between CES and CEB will enable planners to make more informed decisions on how park spaces may influence the resultant CEB enjoyed by users [5]. Figure 10 shows the spatial distribution of CES that relates to a high presence of CEB where CES resulting in high experiential CEB was most widespread and prevalent. This reinforces the finding that the experiential benefits of BAMKP were the most dominant among the three types of CEB as they may be widely enjoyed throughout the park.

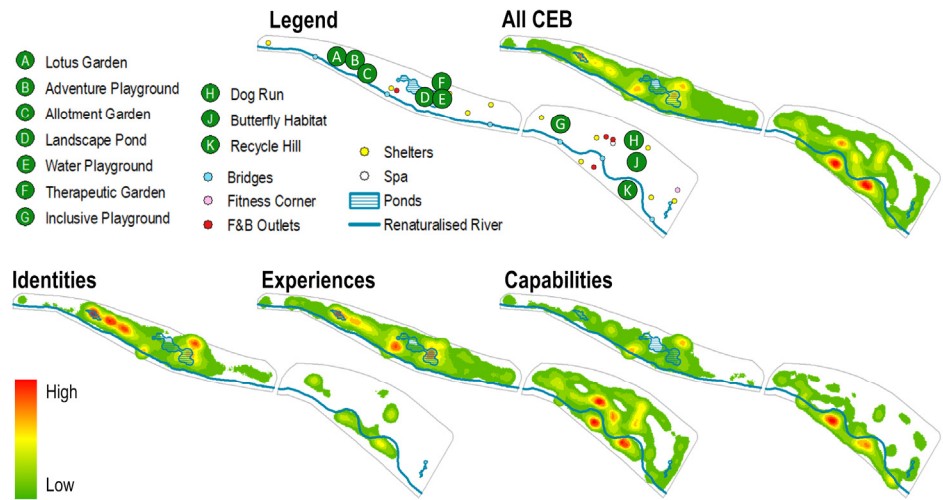

**Figure 10.** Spatial distribution of CES relating to high CEB enjoyment in BAMKP.

Conversely, for the categories of identities and capabilities CEB, the balance is skewed towards the Pond Gardens and slightly towards the River Plains respectively. Most of the CES relating to a high presence of identities CEB were found in the Pond Gardens, with hotspots around its unique key features such as Landscape Pond, Lotus Garden, Adventure Playground and a bridge that provides a scenic view of the naturalised river. As for the capabilities CEB, slightly more CES associated with high presence of capabilities CEB were found in the River Plains located near the McDonald's F&B outlet, Recycle Hill and the similar bridges that provide scenic views of BAMKP.

## 5. Discussion

### 5.1. Policy Implications

Generally, BAMKP has been well received by park users as shown from the generally positive and highly important cultural values attached to the overall park, its features and services. The success of the renaturalised park was evidently shown through the reported wealth of CES present in BAMKP, especially for recreational and aesthetic CES [22]. Congregations of CES hotspots were found near redesigned park features, especially the renaturalised water features [22]. The multiple functions of environmental spaces provided a variety of synergised CES, as suggested by their moderately high spatial correlation. The high presence of CEB enjoyed by park users also compounded its success, most notably for the universally enjoyed experiential CEB via exercising [22].

However, the reported results also suggested the limitations of BAMKP in gearing towards the "City in Nature" vision due to lack in abundance and spatial spread of biodiversity and education CES in BAMKP [13,14,23,24]. These two CES are crucially important to "provide Singaporeans with a better quality of life while co-existing with flora and fauna on this island" [12]. By enhancing the biodiversity in parks, the chances of human-wildlife interactions may be enhanced [3,4,41]. However, it is important for these interactions to be positive as well, which then requires park users to have sufficient environmental knowledge and understanding of the parks' flora and fauna [41]. Currently, as proxied by the Google reviews (see Figure 6), the general public may be lacking awareness of the available educational opportunities afforded by BAMKP. The poor engagement with the important biodiversity and educational CES may hamper BAMKP in progressing to a rewilded park suitable for a "City in Nature" [12].

Nevertheless, education and biodiversity CES was still accorded high importance by park users which supports further augmenting of more human-wildlife interactions in BAMKP. As such, five data-driven suggestions are recommended to improve BAMKP towards becoming a more rewilded park. First, park spaces and features may be redesigned to incorporate wilder and more informative elements for general public usage [12]. The former may likely be achieved with the rewilding plans already slated for BAMKP [14]. As for the latter, while current efforts do exist such as the occasional family tours conducted by organisations [42], increasing the frequency and publicity of these educational opportunities for the general public and the usage of mobile applications with interactive augmented reality elements, which complements the C2C Trail [24], may also be considered.

Second, local planners may consider expanding the blue-green renaturalisation of the park northwards to enrich the diversity of CES offered in synergy with the existing popular recreational CES already present there [22]. By enabling park users to both engage in recreational activities whilst enjoying other CES, the overall park experience and CEB enjoyed would likely be enhanced. Third, planners may consider increasing number of visitors to the western Pond Gardens, which was reportedly less frequented but richer in the crucial biodiversity and education CES [22]. This may be accomplished by improving accessibility to the western park coupled with more F&B outlets like those available in eastern River Plains. The adjacent Bright Hill metro station just started its operation in 2021 [43] coupled with the recent launch of the C2C Trail stretching across both sections of BAMKP may also likely help attract more park visitors too [23,24].

Furthermore, planners may emulate the successful naturalised Pond Garden features yielding high identities CEB for spatially targeted interventions in the poorer northern River Plains. Improvement of accessibility to the park to increase park visit frequency may also help to enhance identities CEB enjoyed. The diversification of CEB would be important to "provide Singaporeans with a better quality of life" [12] with the "City in Nature" vision. Lastly, planners may also examine the feasibility to locate the various abovementioned suggestions at the reported CES coldspots, such as the northeast part of BAMKP to maximise the park spaces available with more densely packed CES and CEB for park users. Park features that were deemed of less importance, such as the Therapeutic Garden, Allotment Garden and Lotus Garden, could be reviewed and refurbished accordingly towards the "City in Nature" vision alongside public sentiments.

Overall, the redefined CES conceptual framework revealed a rich plethora of information ranging from the CES, CEB to cultural values of BAMKP as an UGS. The framework has provided quality and actionable insights to support planners in their UGS management. As the shaping of environmental spaces is often directly under the purview of planners, planners will now understand how their decisions may affect the availability of CES and enjoyment of CEB, consequently allowing them to make more informed decisions for their desired outcomes [3–5]. A reassessment using the framework performed afterwards can also help to evaluate the effectiveness of any changes effected to improve the UGS. This paper, hence, encourages future planners to consider employing the redefined CES

conceptual framework to obtain more relational, data-driven and actionable insights to better support UGS management.

*5.2. Methodological Implications*

Due to the limitations of the online PPGIS surveys and social media text mining, this research would like to encourage future CES research to deliberate on the following significant methodological considerations for more effective UGS management.

First, it is important for future research to continue employing public participatory methods rather than having a singular reliance on the traditional top-down expert opinions of planners and researchers [1–4]. Only by supplementing professional opinions with the ground and bottom-up data obtained from the actual UGS users may a truer reality of the UGS be uncovered for more effective planning and management [19]. This is especially crucial for the employment of CES as a conceptual backbone for research given its strong social dimension where the direct collection of data from the UGS users may reveal a more accurate presence of the services, benefits and valuations of an UGS. Engaging with these key stakeholders and beneficiaries of the UGS should be ideally performed during the research design phase [1–4,19,20], This is an acknowledged limitation of this study as the survey questionnaire design was solely based on the researcher's extensive and frequent personal experiences with BAMKP. Thus, planners should consider collecting public-consulted data for better understanding of the reality on the ground in order to make more informed decisions. This recommendation extends beyond the border of BAMKP and Singapore to other sites as well, highlighting the necessity of science-policy interfaces.

Second, future research should deliberate on the choice of PPGIS tool employed to collect purposeful data. The affordances of Map-Me were tapped upon to allow participants to respond to spatial questions using a "spray can" to add blobs onto a curated survey map [31] (refer to Supporting Information Figure S4). This is advantageous, as people's general concept of space and place tend not to be reduced to simply points, lines and polygons as afforded by traditional PPGIS mapping tools [1–4]. Hence, truer reflections of CES may be obtained via Map-Me. However, Map-Me is slightly limited with its incompatibility with mobile devices to perform the survey. Thus, future research should consider the various advantages and disadvantages of the available PPGIS tools, including financial cost and accessibility of data collection, to meet their research needs.

Third, as "no single methodology can capture the total cultural value of any ecosystem" and the complexity of the relationship between CES, CEB and cultural values in relation to individual people, certain aspects of it might not be fully captured by the blind spots present in every research method [25,44]. Future studies should consider using mixed methods to overcome possible methodological limitations and uncover a closer-to-reality truth of the UGS. For instance, the methods employed in this paper were able to successfully capture the "what" and "where" of the CES but were less able to explain the "how" and "why" for their presence which may be better revealed from interviews with park users. The PPGIS survey obtained from a limited sample size ($n = 100$) with slightly skewed demographics may also be less of a concern due to the validation by similar results from the sizable ($n = 5762$) and more demographically balanced social media text mining of Google reviews. Hence, UGS planners should consider employing complementary mixed methods to augment the reliability of the findings in supporting UGS planning and decision making.

Besides, deliberations should be made between balancing the accessibility and reliability of the data collection process. While it is ideal to augment both factors, in reality, some level of optimisation needs to be performed to calibrate the two seemingly inversely-related factors. For example, the administering of survey questionnaires online may be more convenient for both the planners and participants in terms of time and effort spent [45]. However, the reliability of the data collected online may be likely undermined by potential lower response rate and mapping effort, especially for the spatial PPGIS questions [46,47]. The collection of data online may also potentially skew the demographics of respondents towards the younger age groups. As this may reduce the representativeness and thus

reliability of the findings, intentional sampling effort should be established to ensure a more representative sample.

Lastly, continuous and real-time monitoring of the constantly evolving CES, CEB and cultural values in an UGS may also be useful for urban planners. By receiving timely updates, urban planners may better take mitigating actions that are data-driven to enact changes and improvements to the UGS [48]. As compared to static findings momentarily captured through one-off research, continuous monitoring may reflect truer evolving values in an UGS on a more temporally sensitive and longitudinal scale. Apart from continuous traditional surveying, urban planners can rely on more suitable platforms such as Maptionnaire (https://maptionnaire.com/) which allows real-time spatial data to be gathered from the public and presented in a more communicative manner. Places with greater access to more advanced technology might consider leveraging the Internet of Things with widespread sensors distributed across the UGS for continuous monitoring. Singapore has the potential to achieve this with the Smart Nation initiative where urban planning and management are to be supported with a plethora of information delivered from a shared network of sensors [49]. Other than static sensors, it could also be possible to deploy mobile sensors to collect spatially dynamic data. Singapore has also recently experimented with this via a roving robot dog equipped with video analytic tools to estimate number of visitors in BAMKP [50]. Overall, future studies should carefully deliberate on the abovementioned methodological considerations to collect more credible data for constructive UGS management.

## 6. Conclusions

To encapsulate, this paper has sought to evaluate the operability of Tandarić et al.'s (2020) redefined CES conceptual framework to support UGS planners in their decision making and management, with a specific focus on Singapore's "City in Nature" national initiative. BAMKP was utilised as a case study research area where the redefined CES approach formed the conceptual backbone in methodologically distilling the CES, CEB and cultural values of the park. PPGIS served as the primary method and provided spatial and non-spatial insights regarding the "what" and "where" CES were present in BAMKP, alongside CEB and cultural values. To buttress and better validate the PPGIS findings, social media text mining analysis via Google reviews was also performed through occurrence frequency and sentiment analysis.

The four research questions were addressed. First, it was found that the public's cultural values of BAMKP were generally positive with park users generally valuing most park features and services with high importance. This is especially true with regard to the paths, bridges and naturalised rivers that relate strongly to the recreation CES. Second, it was discovered that BAMKP offered a wealth of CES, especially recreation and aesthetic CES that were most prevalently offered to park users. Third, park users were reported to enjoy a moderately high level CEB with experiential CEB being the most widely enjoyed. Lastly, the relationship between CES and CEB was also uncovered, where it was found that while experiential CEB may have been resulted from CES offered throughout the park, identities CEB and capabilities CEB tend to have been concentrated towards the western Pond Gardens and eastern River Plains respectively.

While this paper has reported BAMKP as a relatively successful renaturalised urban park, the usage of the redefined CES approach has also exposed various shortcomings of the park in gearing towards Singapore's "City in Nature" vision. From the actionable insights obtained, five data-driven recommendations were provided: (i) design park spaces and features with wilder and more informative elements, (ii) expand blue-green naturalisation of the park northwards, (iii) improve accessibility and attractiveness of the western Pond Gardens richer in biodiversity and education CES, (iv) rebalance spread of identities and capabilities CEB by replicating success, and (v) refurbish CES coldspots in conjunction with abovementioned suggestions to maximise park space. These suggestions may better ready BAMKP towards the "City in Nature" vision.

It was evident from this paper that the redefined CES conceptual framework has the potential to be researched further for it to be operationalised to better support planners in decision making and management of UGS. This is because the CES approach doubly decouples CES into its component environmental spaces and cultural activities where their interactions then generate CEB. Planners may draw insights on how shaping of environmental spaces may influence the CES offered, the CEB enjoyed by UGS users and hence may make more data-driven and informed decisions when shaping environmental spaces. As such a redefinition does not fixate the CES, CEB or cultural values to be studied, the flexibility afforded to choose relevant and relatable CES, CEB and cultural values to research further enhances the operability of the redefined CES approach.

Reflexivity was performed during and after the data collection and analysis. When using the redefined CES approach to study an UGS for management purposes, this paper would like to encourage future research to deliberate on five methodological considerations: (i) employ public participatory methods to garner truer reflection of accurate data directly from users of the UGS, (ii) carefully consider on the choice of PPGIS tools based on their affordances, (iii) exercise mixed methods to uncover a diverse spectrum of the same data for closer-to-reality truth of an UGS, (iv) optimise and recalibrate between the accessibility and reliability of the data collection method, and (v) consider the potential and feasibility of continuous monitoring to unravel temporally dynamic and longitudinal insights. Overall, this paper would recommend the employment of the redefined CES conceptual framework to generate relational, data-driven and actionable insights to better support UGS management in the future.

**Supplementary Materials:** The following supporting information can be downloaded at: https:// www.mdpi.com/article/10.3390/su14031499/s1, Figure S1: Photographs of prominent park features in BAMKP; Figure S2: Sociodemographic information of PPGIS participants; Figure S3. Sensitivity analysis of CES relating to high CEB enjoyment in BAMKP; Figure S4. Screenshot of PPGIS Map-Me user interface; Table S1: PPGIS one-way ANOVA test information; Method S1. PPGIS online survey questionnaire.

**Author Contributions:** This work is the final year project of Y.F.K. under the academic advisory of H.H.L. and E.P. Conceptualization, H.H.L. and E.P.; methodology, H.H.L. and Y.F.K.; software, Y.F.K.; validation, H.H.L. and E.P.; data curation and visualization, Y.F.K.; resources: E.P.; original draft preparation, Y.F.K.; writing—review and editing, H.H.L.; funding acquisition, H.H.L. and E.P. All authors have read and agreed to the published version of the manuscript.

**Funding:** The research activities are funded by the National Institute of Education at the Nanyang Technological University (SUG-NAP EP3/19) and the Ministry of Education—Singapore (#Tier1 RT06/19, #Tier1 2021-T1-001-056 and #Tier2MOE-T2EP402A20-0001) acquired by EP. This work is also jointly supported Research Initiation Grant from AIT (SET-2021-R011). Co-author HHL also expresses his appreciation to the International Foundation for Science for supporting this study through its Basic Research Grant Programme (NO. I2-W-6511-1).

**Institutional Review Board Statement:** The study was conducted in accordance with the Declaration of Helsinki, and approved by the Institutional Review Board of National Institute of Education (16 October 2020).

**Informed Consent Statement:** Written informed consent has been obtained from the participants to publish this paper.

**Data Availability Statement:** Not applicable.

**Conflicts of Interest:** The authors declare no conflict of interest.

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
