# Peer review of "Towards a “City in Nature”: Evaluating the Cultural Ecosystem Services Approach Using Online Public Participation GIS to Support Urban Green Space Management"

_sustainability, doi:10.3390/su14031499_

Round 1

Reviewer 1 Report

see attached

Author Response

The authors are appreciative of the reviewer's constructive comments and suggestions. Please find attached our detailed response. We strongly believe that our work significantly benefits from your inputs and has evolved into a much higher quality version of itself. 

Reviewer 2 Report

I congratulate you on a very interesting and useful contribution. The major innovation here is the use of social media to provide quantifiable data on the ecosystem services and benefits of UGS. Your methods are well done and your analysis is very appropriate. I also really value your CEB framework which separates identities, experiences and capabilities as measurable elements. I do recommend significant revisions: 1) there are quite a few typographical errors (e.g.. lines 14 and 15 should be "makes" and "definitions") and awkward English grammar errors (e.g., use "encourage" not "implore" in 3 places). This is easily fixed by a good copy-editing by a fluent English editor. 2) There are a lot of papers out there that now use social media mining in the context of parks and conservation and you could do a more thorough job of citing/mentioning a few more of those. 3) I'm sure you are aware there is almost certainly a big age bias here as younger people are much more likely to use social media and online reviews. I don't see where you have have really discussed the implications of that bias. 

Author Response

(The authors gave the same response as above.)

Round 2

Reviewer 1 Report

The authors have addressed the issues raised by the reviewer. 

One issue of concern that should be addressed prior to publication. The authors are encouraged to explain how the questions used in the 20 question online survey were developed. 

Author Response

The authors appreciate the supportive comment of the reviewer. We have added the following texts to clarify how the questionnaire was developed (page 6 lines 136 - 141) 

"The questionnaire survey was developed based on references made to previous studies related to quantifying and mapping CES, see for instance, Yee et al. 2021 [4] or Loc et al. 2021 [5]. The questionnaire was pre-tested at the site and further improved, especially in terms of wording. The final version of the questionnaire used in this study is included as the supplementary material. "

Reviewer 2 Report

Very nice job of revising and addressing my concerns.

Author Response

The authors appreciate very useful and constructive comments from the reviewer. We believe that our work has greatly benefited from your input. It has become a much-improved version.